# The *dmsEFABGH* operon encodes an essential and modular electron transfer pathway for extracellular iodate reduction by *Shewanella oneidensis* MR-1

Lingyu Hou,[1] Beiling Zheng,[1] Zhou Jiang,[1] Yidan Hu,[1] Liang Shi,[1,2] Yiran Dong,[1,2] Yongguang Jiang[1,3]

**ABSTRACT**  Extracellular iodate reduction by *Shewanella* spp. contributes to iodide generation in the biogeochemical cycling of iodine. However, there is a disagreement on whether *Shewanella* spp. use different extracellular electron transfer pathways with dependence on electron donors in iodate reduction. In this study, a series of gene deletion mutants of *Shewanella oneidensis* MR-1 were created to investigate the roles of *dmsEFABGH*, *mtrCAB,* and *so4357–so4362* operons in iodate reduction. The iodate-reducing activity of the mutants was tested with lactate, formate, and $H_2$ as the sole electron donors, respectively. In the absence of single-*dms* gene, iodate reduction efficiency of the mutants was only 12.9%–84.0% with lactate at 24 hours, 22.1%–85.9% with formate at 20 hours, and 19.6%–57.7% with $H_2$ at 42 hours in comparison to complete reduction by the wild type. Progressive inhibition of iodate reduction was observed when the *dms* homolog from the *so4357–so4362* operon was deleted in the single-*dms* gene mutants. This result revealed complementation of *dmsEFABGH* by *so4357–so4362* at the single-gene level, indicating modularity of the extracellular electron transfer pathway encoded by *dmsEFABGH* operon. Under the conditions of all electron donors, significant inhibition of iodate reduction and accumulation of $H_2O_2$ were detected for Δ*mtrCAB*. Collectively, these results demonstrated that the *dmsEFABGH* operon encodes an essential and modular iodate-reducing pathway without electron donor dependence in *S. oneidensis* MR-1. The *mtrCAB* operon was involved in $H_2O_2$ elimination with all electron donors. The findings in this study improved the understanding of molecular mechanisms underlying extracellular iodate reduction.

**IMPORTANCE**  Iodine is an essential trace element for human and animals. Recent studies revealed the contribution of microbial extracellular reduction of iodate in biogeochemical cycling of iodine. Multiple reduced substances can be utilized by microorganisms as energy source for iodate reduction. However, varied electron transfer pathways were proposed for iodate reduction with different electron donors in the model strain *Shewanella oneidensis* MR-1. Here, through a series of gene deletion and iodate reduction experiments, we discovered that the *dmsEFABGH* operon was essential for iodate reduction with at least three electron donors, including lactate, formate, and $H_2$. The *so4357–so4362* operon was first demonstrated to be capable of complementing the function of *dmsEFABGH* at single-gene level.

**KEYWORDS**  iodate reduction, *Shewanella*, extracellular electron transfer, DMSO reductase, *c*-type cytochrome

Iodine is used as a structural element for the synthesis of thyroxine in the thyroid of human and other vertebrates. Both iodine deficiency and excess iodine uptake can cause thyroid diseases (1, 2). Thus, iodine enrichment in food and drinking water has

Address correspondence to Yongguang Jiang, jiangyg@cug.edu.cn.

The authors declare no conflict of interest.

See the funding table on p. 13.

significant effects on human and animal health (3, 4). Most of the iodine in the earth exists as iodide ($I^-$) and iodate ($IO_3^-$) (5–7). Seaweed is a major natural dietary source of iodine, and it prefers to accumulate $I^-$ but not $IO_3^-$ (8). The concentration of $I^-$ in surface seawater is apparently higher than its theoretical value (5), which is favorable for iodine accumulation in the seaweed. Iodine-rich groundwater was distributed widely worldwide. Drinking high-iodine groundwater caused iodine-induced endemic goiter in several countries around the world (4, 9, 10). Low adsorption and strong mobility of $I^-$ have been proven to be an important mechanism for iodine enrichment in groundwater (11, 12). These researches revealed that $I^-$ is the critical species in the process of both biotic and abiotic iodine enrichment.

Iodate reduction by microorganisms has been found to contribute to the generation of $I^-$ in surface seawater as well as in deep groundwater (13–15). Several genera of bacteria were found to contain strains capable of dissimilatory $IO_3^-$ reduction, including *Desulfovibrio* (16), *Shewanella* (15, 17), *Pseudomonas* (18), *Denitromonas* (14), *Aromatoleum* (19), *Azoarcus* (20), *Methylomirabilis* (21), and one genus related to *Agrobacterium* (22). $IO_3^-$ reduction was dependent on the presence of nitrate in an *Agrobacterium*-related strain DVZ35, implying that $IO_3^-$ might be reduced by nitrate reductase in this bacterium (22). On the contrary, nitrate reductase was not involved in $IO_3^-$ reduction by *Shewanella* (17).

An *idrABP1P2* operon involved in $IO_3^-$ reduction was first identified in *Pseudomonas* sp. strain SCT and was conserved in the isolated $IO_3^-$-respiring microbes (14, 19, 20, 23). It was proposed that $IO_3^-$ is reduced by a complex of IdrABP1P2 proteins in the periplasmic space (23). In contrast to the intracellular reduction of $IO_3^-$ by *Pseudomonas* sp. strain SCT, $IO_3^-$ reduction by *S. oneidensis* MR-1 occurs extracellularly (24). Two operons, including *dmsEFAB* and *mtrCAB*, were demonstrated to be necessary for $IO_3^-$ reduction in *S. oneidensis* MR-1 (24–26). The *dmsEFAB* operon encodes a dimethylsulfoxide (DMSO) reduction pathway (Dms) consisting of a molybdenum enzyme DmsA, an iron-sulfur cluster containing subunit (DmsB), a periplasmic *c*-type cytochrome DmsE, and a membrane-bound protein DmsF. DmsA and DmsB form the DMSO reductase complex anchored by DmsF on the outer membrane (27). DmsE accepts electrons from inner membrane quinol oxidase CymA and transfers them to DmsAB (27). In addition, the *dms* operon also contains *dmsG* and *dmsH* genes, which have not been verified to be involved in $IO_3^-$ reduction (27). The *mtrCAB* operon encodes a metal reduction pathway (Mtr) comprising two outer membrane *c*-type cytochromes, MtrA and MtrC, and an anchor protein MtrB (28). MtrC is a metal reductase located on the outer side of the outer membrane and accepts electrons from MtrA located on the inner side (28). Electrons from inner membrane are transferred to MtrA via CymA and periplasmic cytochromes (29). The genome of *S. oneidensis* MR-1 also contains an *so4357–so4362* operon homologous to *dmsEFABGH* and a *mtrDEF* operon homologous to *mtrCAB* (27, 30). Previous study revealed that MtrF and MtrD can partially complement the metal-reducing activity of MtrC and MtrA, respectively, indicating modularity of the MtrCAB pathway (31). However, the role of *so4357–so4362* in extracellular electron transfer has not been systematically investigated. Deletion of *so4357–so4360* has no adverse effect on the reduction of DMSO and $IO_3^-$, indicating that the *so4357–so4362* gene cluster could not encode an independent electron transfer pathway toward DMSO and $IO_3^-$ (26). The wide distributions of *Shewanella* and other bacteria containing both Dms and Mtr pathways in natural environments imply the potential important contribution of microbial extracellular $IO_3^-$ reduction to biogeochemical cycling of iodine (32).

Recently, two different reaction models were proposed for $IO_3^-$ reduction by *S. oneidensis* MR-1 (25, 26). The first is an electron donor-dependent model in which

the Dms pathway and a hybrid electron transfer pathway MtrAB-DmsAB reduce $IO_3^-$ with formate and lactate as the sole electron donors, respectively (24, 25). However, the interaction between MtrAB and DmsAB has not been proven in previous studies. The second is a coordination model in which $IO_3^-$ is reduced into HIO and $H_2O_2$ by the Dms pathway, and $H_2O_2$ is degraded by the Mtr pathway (26). The validity of this model is limited to $IO_3^-$ reduction with lactate as the sole electron donor (26). In natural environments, multiple reduced substances such as lactate, formate, and $H_2$ can be utilized as energy source for microbial growth and survival (33–36). To solve the disagreement in the reaction mechanism, the roles of *dmsEFABGH*, *mtrCAB,* and *so4357–so4362* in iodate reduction were systematically investigated in this study. A series of gene deletion mutants of *S. oneidensis* MR-1 were created, and the $IO_3^-$-reducing activity of the mutants was characterized using lactate, formate, and $H_2$ as the sole electron donors, respectively. The results of this study demonstrated that the Dms pathway was essential for $IO_3^-$ reduction with all electron donors, and *so4357–so4362* could partially complement the function of *dmsEFABGH* at single-gene level. The Mtr pathway was necessary for maximum reduction of $IO_3^-$ and was involved in $H_2O_2$ degradation with all electron donors.

## MATERIALS AND METHODS

### Bacterial strains and culture conditions

*S. oneidensis* MR-1 and *Escherichia coli* (WM3064 and DH5α) were routinely grown in lysogeny broth (LB) medium at 30°C and 37°C, respectively. The 2,6-diaminopimelic acid was supplemented in the medium of *E. coli* WM3064 with a final concentration of 100 µg/mL. Antibiotic-resistant bacterial strains were grown in LB medium with 15 µg/mL gentamicin or 50 µg/mL kanamycin. Sodium salts of lactate, formate, and $IO_3^-$ were used as electron donors or acceptors. If not stated otherwise, all the chemicals used in this study were of analytical grade and purchased from Sigma Aldrich (St. Louis, MO, USA). Table 1 lists all the bacterial strains used in this study.

### Vector construction, gene deletion, and complement

A suicide vector pDS3.2 was constructed by inserting a short sequence with additional SbfI and AvrII sites in plasmid pDS3.0 (38). Two oligonucleotides, sbavF and sbavR (Table S1), were synthesized and mixed at a molar ratio of 1:1. The mixture was denatured at 94°C for 10 s and annealed by natural cooling at room temperature to form a double-strand DNA product with 5′ overhangs. Afterward, the DNA product was diluted and ligated with linear plasmid pDS3.0 generated by digestion with SphI and SacI. The ligation product was vector pDS3.2.

Gene deletion was performed according to the two-step selection strategy described previously (38, 40). The upstream and downstream sequences of each gene were amplified and used to create a fused fragment by overlap extension PCR. The fused DNA fragments were cloned into vector pDS3.2. The recombinant suicide vectors were transformed into *E. coli* WM3064 and introduced into *S. oneidensis* MR-1 by conjugation (41). After selection by gentamicin and sucrose sequentially, positive mutant clones were screened by PCR amplification and sequencing. The PCR primers used for mutant construction were listed in Table S1.

To complement the mutants, the coding sequence of each gene was amplified by using primers with ribosomal binding sites that have been shown to be effective in *S. oneidensis* MR-1 (42, 43) (Table S1). The PCR products were cloned into a broad host range expression vector pBBR1MCS-2 (39). The resultant constructs were transformed into corresponding gene deletion mutants by electroporation. The pulse settings were 25 µF, 400 Ω, and 0.75 kV on a Bio-Rad MicroPulser. Empty vector was transformed into the mutants and the wild-type (WT) strain as controls.

**TABLE 1** Bacterial strains and plasmids used in this study

| Strains or plasmids | Relevant genotype or uses | Source or reference |
|---|---|---|
| Strains | | |
| *S. oneidensis* | | |
| MR-1 | Wild-type *S. oneidensis* | (37) |
| Δ*dmsA* | *dmsA* deletion mutant of *S. oneidensis* MR-1 | This study |
| Δ*dmsB* | *dmsB* deletion mutant of *S. oneidensis* MR-1 | This study |
| Δ*dmsE* | *dmsE* deletion mutant of *S. oneidensis* MR-1 | This study |
| Δ*dmsF* | *dmsF* deletion mutant of *S. oneidensis* MR-1 | This study |
| Δ*dmsG* | *dmsG* deletion mutant of *S. oneidensis* MR-1 | This study |
| Δ*dmsH* | *dmsH* deletion mutant of *S. oneidensis* MR-1 | This study |
| Δ*mtrCAB* | *mtrCAB* deletion mutant of *S. oneidensis* MR-1 | (26) |
| Δ*dmsA*Δ*so4358* | *dmsA* and *so4358* deletion mutant of *S. oneidensis* MR-1 | This study |
| Δ*dmsB*Δ*so4357* | *dmsB* and *so4357* deletion mutant of *S. oneidensis* MR-1 | This study |
| Δ*dmsE*Δ*so4360* | *dmsE* and *so4360* deletion mutant of *S. oneidensis* MR-1 | This study |
| Δ*dmsE*Δ*cctA* | *dmsE* and *cctA* deletion mutant of *S. oneidensis* MR-1 | This study |
| Δ*dmsE*Δ*fccA* | *dmsE* and *fccA* deletion mutant of *S. oneidensis* MR-1 | This study |
| Δ*dmsF*Δ*so4359* | *dmsF* and *so4359* deletion mutant of *S. oneidensis* MR-1 | This study |
| Δ*dmsG*Δ*so4362* | *dmsG* and *so4362* deletion mutant of *S. oneidensis* MR-1 | This study |
| Δ*dmsH*Δ*so4361* | *dmsH* and *so4361* deletion mutant of *S. oneidensis* MR-1 | This study |
| *E. coli* | | |
| WM3064 | Donor strain for conjugation, Δ*dapA* | Lab stock |
| DH5α | *E. coli* host for cloning | Takara Co., Ltd. |
| Plasmids | | |
| pDS3.0 | Suicide vector, Gm$^r$ *sacB* | (38) |
| pDS3.2 | Suicide vector, Gm$^r$ *sacB*, with SbfI and AvrII sites | This study |
| pBBR1MCS-2 | Expression vector, Km$^r$ P$_{lac}$ | (39) |
| pDmsA | Recombinant pBBR1MCS-2 containing *dmsA* coding sequence | This study |
| pDmsB | Recombinant pBBR1MCS-2 containing *dmsB* coding sequence | This study |
| pDmsE | Recombinant pBBR1MCS-2 containing *dmsE* coding sequence | This study |
| pDmsF | Recombinant pBBR1MCS-2 containing *dmsF* coding sequence | This study |
| pDmsG | Recombinant pBBR1MCS-2 containing *dmsG* coding sequence | This study |
| pDmsH | Recombinant pBBR1MCS-2 containing *dmsH* coding sequence | This study |
| pSO4357 | Recombinant pBBR1MCS-2 containing *so4357* coding sequence | This study |
| pSO4358 | Recombinant pBBR1MCS-2 containing *so4358* coding sequence | This study |
| pSO4359 | Recombinant pBBR1MCS-2 containing *so4359* coding sequence | This study |
| pSO4360 | Recombinant pBBR1MCS-2 containing *so4360* coding sequence | This study |
| pSO4361 | Recombinant pBBR1MCS-2 containing *so4361* coding sequence | This study |
| pSO4362 | Recombinant pBBR1MCS-2 containing *so4362* coding sequence | This study |

## Growth measurement

The mutants and wild-type *S. oneidensis* MR-1 were inoculated in culture tubes containing 8 mL LB medium with an initial optical density of 0.1 at 600 nm ($OD_{600} = 0.1$). The tubes were incubated at 30°C with a shaking speed of 120 rpm. At predetermined time points, the $OD_{600}$ of cultures was measured to obtain their aerobic growth curves.

Anoxic growth of the bacterial strains was tested in modified M1 medium containing 30.00 mM PIPES, 7.50 mM NaOH, 28.04 mM $NH_4Cl$, 1.34 mM KCl, 4.35 mM $NaH_2PO_4$, and 1.50 mM $Na_2SO_4$ supplemented with trace amounts of minerals, vitamins, and amino acids (44–46). The medium was also supplemented with 20 mM lactate and 20 mM fumarate as the sole electron donor and the sole terminal electron acceptor, respectively. To eliminate dissolved oxygen, prepared medium was boiled and flushed for 20 min with pure nitrogen gas before being sealed with thick butyl rubber stoppers. The strains were precultured aerobically in LB medium. Bacterial cells were harvested by centrifugation and washed twice using anoxic medium. Afterward, the cells were inoculated in

anaerobic tubes with disposable syringe. Each tube containing 15 mL anoxic medium and the initial $OD_{600}$ of cultures was approximately 0.1. The tubes were incubated at 30°C with gentle shaking. At predetermined time points, the $OD_{600}$ of cultures was measured to obtain their anoxic growth curves. Bacterial washing and inoculation procedures were performed in an anaerobic chamber (Coy Laboratory Products, USA). During bacterial collection, the mixed gases ($N_2$:$CO_2$:$H_2$ = 95.5:3:1.5) in the chamber were replaced by pure nitrogen gas.

## $IO_3^-$ reductions

$IO_3^-$ reduction activity of the bacterial strains was tested in anoxic M1 medium with 20 mM lactate and 250 μM $IO_3^-$ as the sole electron donor and terminal electron acceptor, respectively (24, 26). The strains were precultured aerobically in LB medium, and bacterial cells were harvested by centrifugation at an $OD_{600}$ of 1.2. The cell pellets were washed twice using anoxic M1 medium and inoculated in anaerobic bottles containing 50 mL anoxic M1 medium. The $OD_{600}$ of the starting cultures was 0.15 if not specified. Negative controls were set by adding the same quantity of wild-type cells killed by boiling in the medium. In addition to lactate, formate and hydrogen gas were also used as sole electron donors. For formate test, 10 mM of formate was supplemented in the M1 medium. For $H_2$ test, 15 mL anoxic M1 medium was prepared in anaerobic tubes and sparged for 5 min with mixed $N_2$:$H_2$ (70:30) gas before being sealed. An $IO_3^-$ reduction test without adding any electron donors was also performed.

At predetermined time points, the concentration of $IO_3^-$ in the medium was measured with the $IO_3^-$-triiodide method (17, 47). Briefly, a 600 μL volume of culture was taken and filtered with a 0.22-μm syringe filter (Jinteng, China). Then, 200 μL filtrate was sequentially mixed with 400 μL sodium citrate buffer (0.1 M, pH 3.3), 1 mL Millipore water, and 400 μL potassium $I^-$ solution (75 mM). Within 3 min, the absorbance of the reaction solutions was measured at 352 nm using a UV spectrophotometer (Thermo Scientific, USA). The concentrations of $IO_3^-$ were calibrated according to a standard curve generated by using $IO_3^-$ solutions with defined concentrations.

## DMSO reductions

Bacterial cells were precultured and harvested as described for $IO_3^-$ reduction test. The cell pellets were washed twice using anoxic LB medium and inoculated in anaerobic tubes containing 15 mL LB medium with 50 mM DMSO as the sole terminal electron acceptor and 40 mM lactate as an additional electron donor (26). The initial $OD_{600}$ of the cultures was 0.1. At predetermined time points, the $OD_{600}$ of the cultures was measured, and growth curves were generated to evaluate the DMSO reduction activity of the strains. Preparation of anoxic LB medium was the same as that of anoxic M1 medium.

## $H_2O_2$ measurement

$H_2O_2$ was measured according to a previously described method in which 2,2′-azino-bis(3-ethylbenzothiazoline-6-sulfate) (ABTS) was oxidized into $ABTS^{·+}$ by $H_2O_2$ in the presence of horseradish peroxidase (HRP) as a catalyst (26, 48). $IO_3^-$ reduction reactions were implemented as aforementioned except that M1 medium was replaced by phosphate-buffered saline solution (PBS). Before inoculation of bacterial cells, ABTS and HRP were added into PBS with final concentrations of 1 mM and 5 μg/mL, respectively. A volume of 200 μL sample was taken and filtered every 5 min. The absorbance of filtrate was measured at 405 nm within 5 min by a Microplate Reader (SpectraMax 190, Molecular Devices, USA). The concentrations of $H_2O_2$ were calibrated according to a standard curve generated by using $H_2O_2$ solutions with defined concentrations.

## Statistical analysis

All experiments in this study were performed in biological triplicate. The $t$-test was used to evaluate the significance of differences in the efficiency of $IO_3^-$ reduction, the optical density, and the $H_2O_2$ contents between bacterial strains.

## RESULTS AND DISCUSSION

### $IO_3^-$ reduction with lactate

Compared to the wild type, all the *dms* mutants showed slower reduction of $IO_3^-$ with lactate as the sole electron donor (Fig. 1A). The reduction rate decreased with an order of WT > Δ*dmsE* > Δ*dmsF* > Δ*dmsH* > Δ*dmsG* ≈ Δ*dmsA* ≈ Δ*dmsB*. At 24 hours, WT reduced all the added 250 µM $IO_3^-$, and no reduction was observed in the killed-cells control. However, Δ*dmsA*, Δ*dmsB*, and Δ*dmsG* only reduced 12.9%–16.3% of the added $IO_3^-$, and Δ*dmsE*, Δ*dmsF*, and Δ*dmsH* reduced 84.0%, 66.4%, and 35.6%, respectively (Fig. 1B). Similarly, severe inhibition of $IO_3^-$ reduction by Δ*dmsB* was also reported in a previous study with lactate and formate as the electron donors (25). To exclude any polar effect of gene deletion, the mutants were complemented with expression vectors carrying *dms* genes. The empty vector served as a negative control. Addition of vectors containing respective *dms* genes in the mutants restored their $IO_3^-$ reduction activity to a level

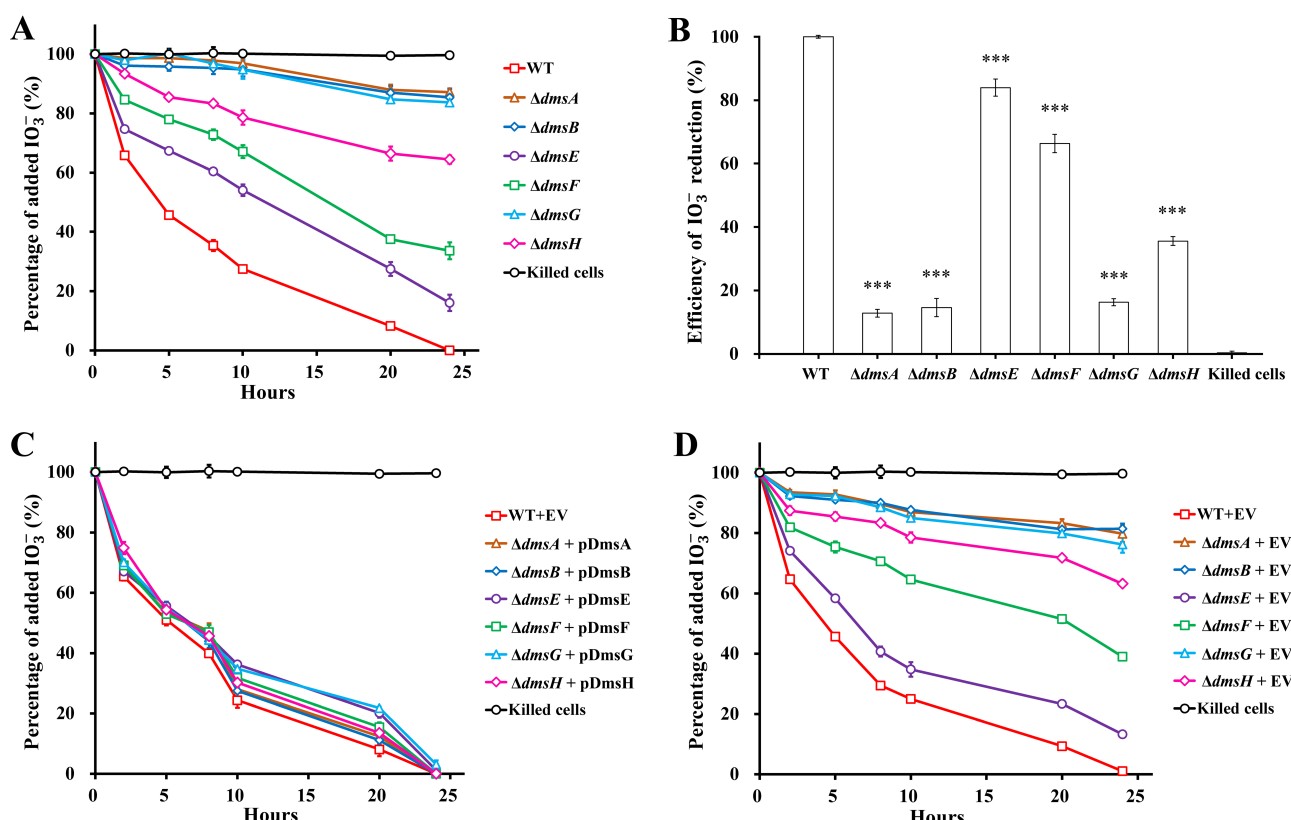

**FIG 1** The roles of *dmsEFABGH* in $IO_3^-$ reduction with lactate as the sole electron donor. Percentage of added $IO_3^-$ over 24 hours of reduction by (A) *dmsEFABGH* deletion mutants, (C) complementation strains, and (D) deletion mutants with empty vector. (B) Efficiency of $IO_3^-$ reduction by *dmsEFABGH* deletion mutants at 24 hours. WT, *S. oneidensis* MR-1; EV, empty vector pBBR1MCS-2. One hundred percent of added $IO_3^-$ was equal to 250 µM. The values reported are the means and standard deviations of triplicate experiments. For points without error bar, the error was smaller than the symbol. Asterisks, significance levels of difference between the mutants and WT, $0.01 < P < 0.05$ (*), $0.001 < P < 0.01$ (**), $P < 0.001$ (***).

similar to that of the wild type (Fig. 1C). Moreover, addition of empty vector did not improve the reduction activity of the mutants (Fig. 1D). Therefore, polar effects were not detected in any mutant tested. In addition, $IO_3^-$ reduction was severely inhibited for both wild-type and mutant strains when electron donors were omitted, indicating that exogenous electron donors were necessary for $IO_3^-$ reduction by *S. oneidensis* MR-1 (Fig. S1).

The *dms* gene cluster was known to encode an electron transfer pathway for extracellular DMSO reduction in *S. oneidensis* MR-1 (27). To confirm the effects of *dms* gene deletion on the function of this pathway, the growth of each *dms* mutant was tested with DMSO as the sole terminal electron acceptor. All the mutants grew much slower than the wild type with a decreasing order of WT > Δ*dmsE* > Δ*dmsF* ≈ Δ*dmsH* > Δ*dmsG* ≈ Δ*dmsA* ≈ Δ*dmsB* (Fig. S2A). After 22 hours of incubation, the optical density of Δ*dmsA*, Δ*dmsB*, Δ*dmsE*, Δ*dmsF*, Δ*dmsG*, and Δ*dmsH* was only 0.21 ± 0.01, 0.17 ± 0.01, 0.51 ± 0.03, 0.26 ± 0.01, 0.20 ± 0.01, and 0.25 ± 0.01, in comparison to 1.26 ± 0.04 for the wild type (Fig. S2B). This result was consistent with previous results that the deletion of *dmsEFAB* genes impaired DMSO reduction (27, 31). However, the function of *dmsG* and *dmsH* has so far not been verified. This study provided direct evidence for the participation of *dmsGH* in both reductions of DMSO and $IO_3^-$. Conserved domain analysis indicated that DmsG encodes a TorD-like chaperone (27). TorD is involved in the maturation of the molybdoenzyme TorA to make it competent to receive the bis(molybdopterin guanine dinucleotide)molybdenum cofactor (49). Therefore, DmsG probably participates in the maturation of the molybdoenzyme DmsA. Although the function of DmsH is still unknown, it was predicted to be a cytoplasmic protein like DmsG (50). In addition, the growth of the *dms* gene mutants was also tested with $O_2$ and fumarate as the sole terminal electron acceptors, respectively. The growth activity of the mutants was close to that of wild-type *S. oneidensis* MR-1 (Fig. S3), confirming that deletion of any *dms* gene had no adverse effects on the reduction of $O_2$ and fumarate (26, 27).

## $IO_3^-$ reduction with formate and $H_2$

Under the conditions of formate and $H_2$ as the sole electron donors, all the *dms* mutants showed slower reduction of $IO_3^-$ than wild-type *S. oneidensis* MR-1 (Fig. 2A and C). The reduction rate decreased with an order of WT > Δ*dmsE* > Δ*dmsF* > Δ*dmsH* > Δ*dmsG* ≥ Δ*dmsA* > Δ*dmsB*, consistent with that observed with lactate. All added $IO_3^-$ was reduced by the wild type at 20 and 42 hours with formate and $H_2$ as the electron donors, respectively (Fig. 2). However, at 20 hours after reduction with formate, Δ*dmsA*, Δ*dmsB*, and Δ*dmsG* only reduced 19.6%–21.7% of the added $IO_3^-$, and Δ*dms*E, Δ*dms*F, and Δ*dms*H reduced 57.7%, 41.2%, and 26.7%, respectively (Fig. 2B). At 42 hours after reduction with $H_2$, Δ*dmsA*, Δ*dmsB*, and Δ*dmsG* only reduced 22.1%–28.0% of the added $IO_3^-$, and Δ*dms*E, Δ*dms*F, and Δ*dms*H reduced 85.9%, 44.8%, and 36.4%, respectively (Fig. 2D). No $IO_3^-$ reduction was observed in the killed-cells control. These results indicated that all *dmsEFABGH* genes are essential for maximum $IO_3^-$ reduction by *S. oneidensis* MR-1 with each of the three electron donors such as lactate, formate, and $H_2$.

## Roles of *so4357–so4362* in $IO_3^-$ reduction

As displayed in Fig. 1 and 2, the *dms* mutants still had partial capability of $IO_3^-$ reduction, indicating the existence of complementary components for *dms* genes in *S. oneidensis* MR-1. Each *dms* gene has a homolog in the *so4357–so4362* gene cluster (27). However, reduction activity for $IO_3^-$ or DMSO was not observed for the gene cluster *so4357–so4360* in a mutant lacking *dmsEFAB* (26). A previous study reported that overexpression of *so4359* and *so4360* can restore the ferric citrate reduction activity of *mtr* mutants (51). This finding inspired us that single genes of *so4357–so4360* may be functional in $IO_3^-$

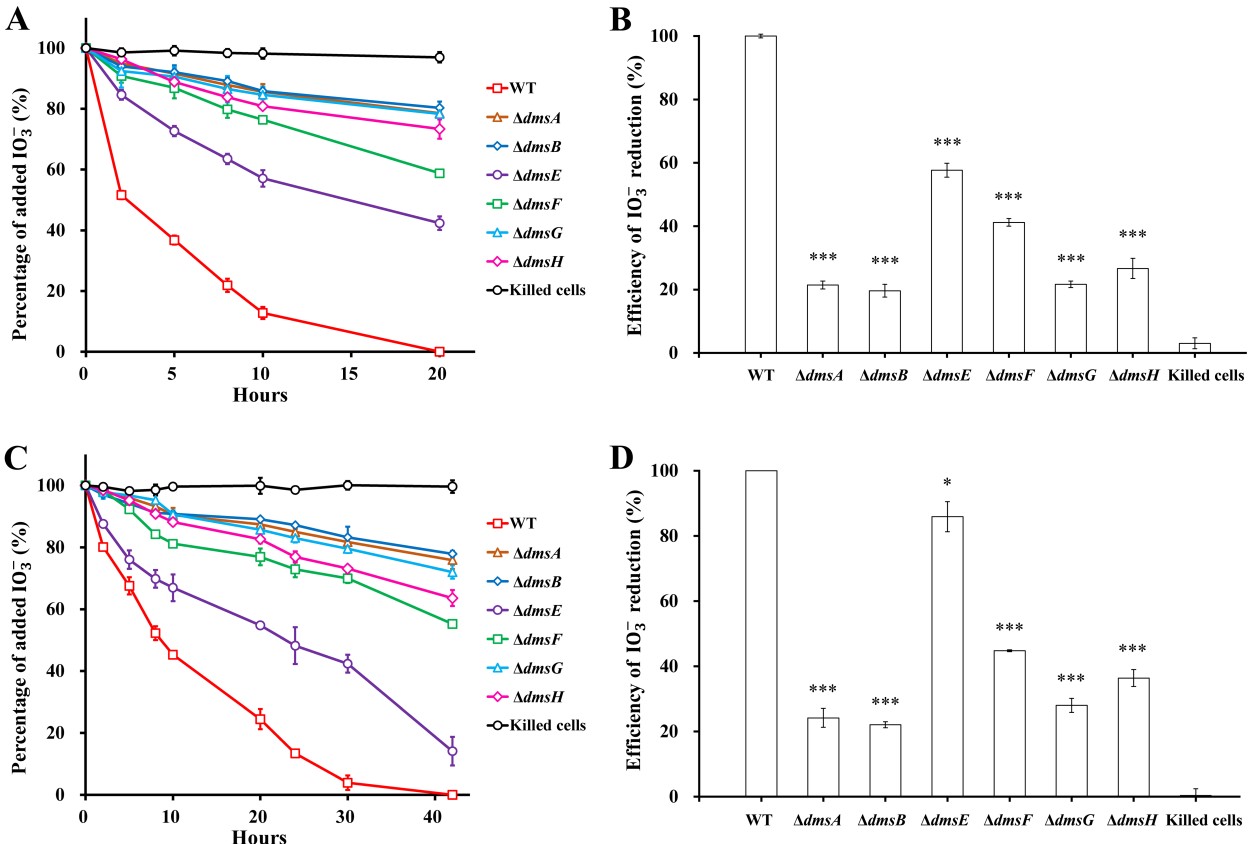

**FIG 2** The roles of *dmsEFABGH* in $IO_3^-$ reduction with formate or $H_2$ as the sole electron donor. (A) Percentage of added $IO_3^-$ over 20 hours of reduction and (B) efficiency of $IO_3^-$ reduction at 20 hours with formate as the sole electron donor. (C) Percentage of added $IO_3^-$ over 42 hours of reduction and (D) efficiency of $IO_3^-$ reduction at 42 hours with $H_2$ as the sole electron donor. WT, *S. oneidensis* MR-1. One hundred percent of added $IO_3^-$ was equal to 250 µM. The values reported are the means and standard deviations of triplicate experiments. For points without error bar, the error was smaller than the symbol. Asterisks, significance levels of difference between the mutants and WT, $0.01 < P < 0.05$ (*), $0.001 < P < 0.01$ (**), $P < 0.001$ (***).

reduction. To verify this speculation, double-gene mutants with simultaneous loss of *dms* gene and its homolog were generated by selectively deleting each gene of *so4357–so4362* in the *dms* mutants (Table 1). Double-gene deletions resulted in progressively slower $IO_3^-$ reduction than the absence of only *dms* gene (Fig. 3). Note that the difference was significant between double-gene mutants and single-*dms*-gene mutant except $\Delta dmsA\Delta so4358$ and $\Delta dmsA$. The results demonstrated that each of the *so4357–so4362* genes could partially complement the function of its *dms* homolog in $IO_3^-$ reduction, although *so4357–so4362* could not encode an active $IO_3^-$ reduction pathway by itself (26). Moreover, this finding showed modularity of the electron transfer pathway encoded by *dmsEFABGH*. In contrast to the results of $IO_3^-$ reduction, similar growth activity was observed between the double-gene mutants and the single-gene mutants with DMSO as the sole terminal electron acceptor (Fig. S4). This result indicated that the genes of *so4357–so4362* could not complement the function of their *dms* homologs in DMSO reduction.

To exclude any polar effect of gene deletion, the double-gene mutants were complemented with expression vectors carrying genes from the *so4357–so4362* operon. The empty vector served as a negative control. Addition of vectors containing *so4357–so4362* genes in the mutants partially restored their $IO_3^-$ reduction activity (Fig. S5).

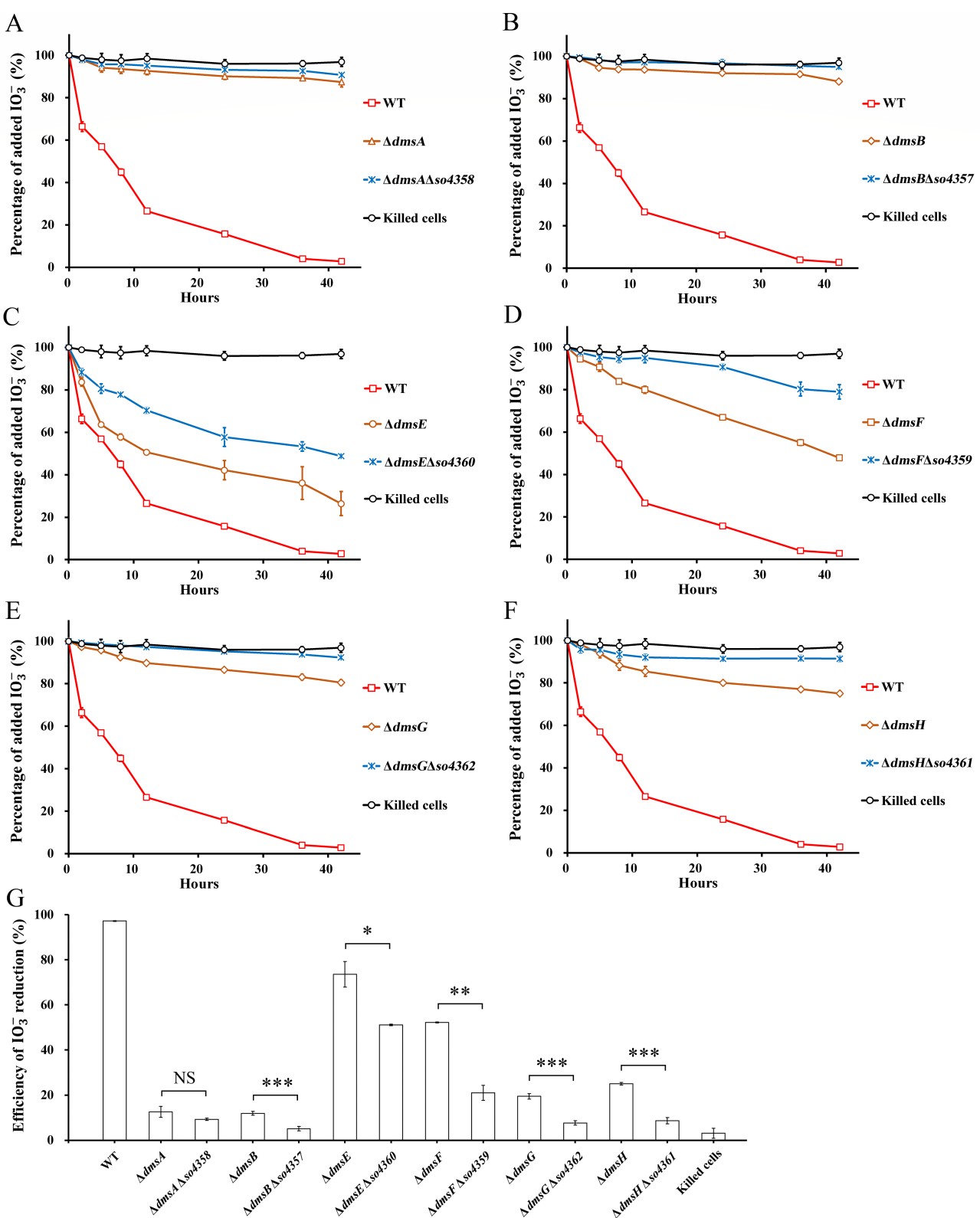

**FIG 3** The roles of *so4357–so4362* in $IO_3^-$ reduction with lactate as the sole electron donor. (A–F) Percentage of added $IO_3^-$ over 42 hours of reduction by *dmsEFABGH* mutants and double-gene mutants. (G) Efficiency of $IO_3^-$ reduction at 42 hours. WT, *S. oneidensis* MR-1. One hundred percent of added $IO_3^-$ was equal to 250 µM. The values reported are the means and standard deviations of triplicate experiments. For points without error bar, the error was smaller than the symbol. The $OD_{600}$ of the starting cultures was 0.1. Asterisks, significance levels of difference between the double-gene mutants and the single-gene mutants, $0.01 < P < 0.05$ (*), $0.001 < P < 0.01$ (**), $P < 0.001$ (***). NS, not significant.

However, addition of empty vector did not improve the reduction activity of the mutants. Therefore, polar effects were not detected in any double-gene mutant tested.

As shown in Fig. 3G, $\Delta dmsE\Delta so4360$ eliminated 51.2% of total $IO_3^-$ after 42 hours of reduction, in comparison to the severely inhibited $IO_3^-$ reduction by $\Delta dmsA\Delta so4358$ (9.3%), $\Delta dmsB\Delta so4357$ (5.1%), $\Delta dmsF\Delta so4359$ (21.0%), $\Delta dmsG\Delta so4362$ (7.7%), and $\Delta dmsH\Delta so4361$ (8.6%). This result is consistent with the finding that the $IO_3^-$-reducing efficiency of $\Delta dmsE$ is higher than those of other single-gene mutants. In previous study, partial defect of $\Delta dmsE$ in DMSO reduction was ascribed to the induced overexpression of $dmsFAB$ (27), which may also be a reason for the weak inhibition of $IO_3^-$ reduction by $\Delta dmsE\Delta so4360$ as well as $\Delta dmsE$. This compensation effect depends on periplasmic electron carriers that could complement the function of DmsE. MtrA and MtrD are both homologs of DmsE (52, 53). However, these two Mtr cytochromes cannot transfer electrons to DmsB (31). Therefore, in addition to DmsE and SO4360, some unknown periplasmic electron carriers were also involved in electron transfer to DmsB.

The tetraheme cytochromes FccA (also known as Fcc3) and CctA (also known as STC) are the major constituents of the periplasmic cytochrome pool (54, 55). As displayed in Fig. S6A, neither $\Delta dmsE\Delta cctA$ nor $\Delta dmsE\Delta fccA$ showed any additive effects on $IO_3^-$ reduction in comparison to $\Delta dmsE$, indicating that the function of $dmsE$ cannot be complemented by $cctA$ or $fccA$. In contrast, a previous study proposed that CctA is able to facilitate electron transfer from CymA to DmsB in the absence of DmsE according to the finding that $\Delta dmsE\Delta cctA$ grew a little slower than $\Delta dmsE$ with DMSO as the sole terminal electron acceptor (31). Growth test in this study also revealed that $\Delta dmsE\Delta cctA$ grew slower than $\Delta dmsE$ and $\Delta dmsE\Delta fccA$ with DMSO (Fig. S6B). Moreover, recent study found that $\Delta fccA\Delta cctA$ completely lost growth activity with DMSO, although deletion of $fccA$ or $cctA$ only caused partial defects (56). These findings revealed the critical and overlapping roles of FccA and CctA in mediating electron transfer from CymA to DmsE (56).

Growth of the double-gene mutants was also tested by using $O_2$ and fumarate as the sole terminal electron acceptors, respectively (Fig. S7). Under aerobic conditions, the growth activity of the mutants was similar to that of wild-type *S. oneidensis* MR-1. With fumarate, $\Delta dmsE\Delta fccA$ had a lower growth rate than the wild-type and other mutants because $fccA$ encodes a fumarate reductase. Therefore, deletion of $dms$ and $so4357$–$so4360$ as well as $cctA$ has no adverse effects on the use of $O_2$ and fumarate.

## Roles of $mtrCAB$ in $IO_3^-$ reduction

The inhibition effects of $dmsEF$ deletions on $IO_3^-$ reduction with lactate were inconsistent with the previously suggested lactate-dependent MtrAB-DmsAB model (25). In reference (25), deletion of $mtrA$ severely impaired $IO_3^-$ reduction with lactate, and further deletion of $dmsEF$ had no additive effects. However, another study found that the deletion of $mtrCAB$ only partially inhibited the reduction of $IO_3^-$ with lactate, and MtrCAB was involved in the degradation of $H_2O_2$ generated as an intermediate product of $IO_3^-$ reduction (26). In this study, the $IO_3^-$ reduction by $\Delta mtrCAB$ was further tested with three electron donors. The $\Delta mtrCAB$ strain used in this study was constructed and verified in a previous study (26). In all tests, $\Delta mtrCAB$ showed obviously slower rates of $IO_3^-$ reduction than the wild type (Fig. 4A). After 36 hours of reduction, WT eliminated nearly all the added $IO_3^-$, and no reduction was observed in the killed-cells control. However, the reduction efficiency of $\Delta mtrCAB$ was only 38.5%, 72.6%, and 68.5% with lactate, formate, and $H_2$ as the sole electron donors, respectively (Fig. 4B). This finding demonstrated that $mtrCAB$ was involved in $IO_3^-$ reduction by *S. oneidensis* MR-1 with the three electron donors tested.

Accumulation of $H_2O_2$ was observed during $IO_3^-$ reduction by $\Delta mtrCAB$ (Fig. 4C), and the maximum concentration of $H_2O_2$ reached 3.80 ± 0.15, 6.31 ± 0.25, and 2.91 ±

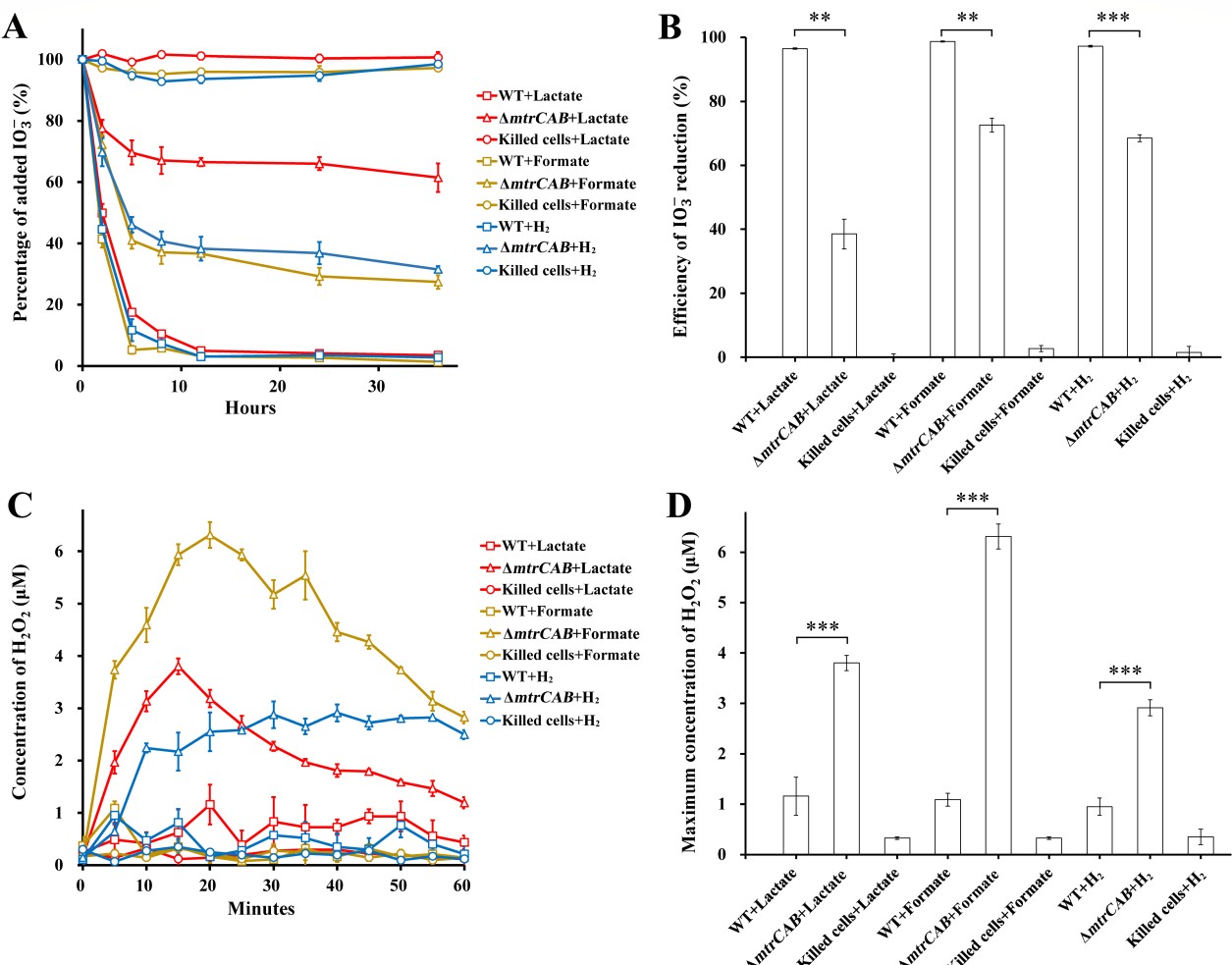

**FIG 4** The roles of *mtrCAB* in $IO_3^-$ reduction with lactate, formate, or $H_2$ as the sole electron donor. (A) Percentage of added $IO_3^-$ over 36 hours of reduction by Δ*mtrCAB*. (B) Efficiency of $IO_3^-$ reduction at 36 hours. (C) Concentration of $H_2O_2$ over 60 min of $IO_3^-$ reduction and (D) its maximum values. WT, *S. oneidensis* MR-1. One hundred percent of added $IO_3^-$ was equal to 250 µM. The values reported are the means and standard deviations of triplicate experiments. For points without error bar, the error was smaller than the symbol. Asterisks, significance levels of difference between Δ*mtrCAB* and WT, $0.01 < P < 0.05$ (*), $0.001 < P < 0.01$ (**), $P < 0.001$ (***).

0.16 µM with lactate, formate, and $H_2$ as the sole electron donors, respectively (Fig. 4D). In contrast to Δ*mtrCAB*, wild-type *S. oneidensis* MR-1 generated less than $1.16 \pm 0.38$ µM $H_2O_2$ with any electron donor (Fig. 4C and D). These findings further demonstrated that *mtrCAB* plays a role in $H_2O_2$ degradation with all the tested electron donors (26). In addition, previous study revealed that multiple enzymes, including catalase, peroxidase, and alkylhydroperoxide reductase, were involved in defense against $H_2O_2$ in *S. oneidensis* MR-1 (57, 58). Presence of these enzymes in Δ*mtrCAB* probably caused the progressive degradation of the accumulated $H_2O_2$ with lactate and formate (Fig. 4C). The absence of $H_2O_2$ degradation by Δ*mtrCAB* with $H_2$ may be ascribed to the lack of carbon source.

Results from this study demonstrated that the $IO_3^-$-reducing mechanism of *S. oneidensis* MR-1 was independent on electron donors. This kind of independence may also be explained by the catabolic mechanisms of electron donors in *S. oneidensis* MR-1. In the case of lactate, a D-lactate dehydrogenase Dld-II and an L-lactate dehydrogenase complex LldEGF were used for its oxidation (59). Dld-II was found to contain a 4Fe-4S-binding domain implementing electron transfer function by using membrane-associated

quinones (e.g., menaquinone) as electron-accepting cofactors (60, 61). The LldF subunit of LldEGF complex also contains a 4Fe-4S-binding domain, indicating an electron transfer role like Dld-II (59). Multiple dehydrogenase complexes were involved in formate oxidation, and menaquinone was also used as an electron-accepting cofactor in the reaction (34, 62, 63). For $H_2$ oxidation, an [Ni-Fe] hydrogenase complex HyaABC was demonstrated to be essential (64). Moreover, the HyaC (previously denoted as HydC) subunit from *Wolinella succinogenes* was proven to be an electron transfer mediator from $H_2$ to quinone (65). In addition, an [Fe-Fe] hydrogenase complex HydABC was also found to have $H_2$-oxidizing activity, and HydC was a menaquinone-binding subunit (63, 66). Therefore, membrane-bound quinones (e.g., menaquinone) were probably common electron sinks for lactate, formate, and $H_2$. Under anaerobic respiration conditions, electrons from menaquinone were transferred to periplasmic cytochromes or oxidoreductases by the inner membrane quinol oxidase CymA (55, 67). Deletion of CymA caused severe impairment of $IO_3^-$ reduction with all electron donors including lactate, formate, and $H_2$ (24). Taken together, these findings suggest that electron donors are not likely to change the electron transfer pathways for extracellular $IO_3^-$ reduction.

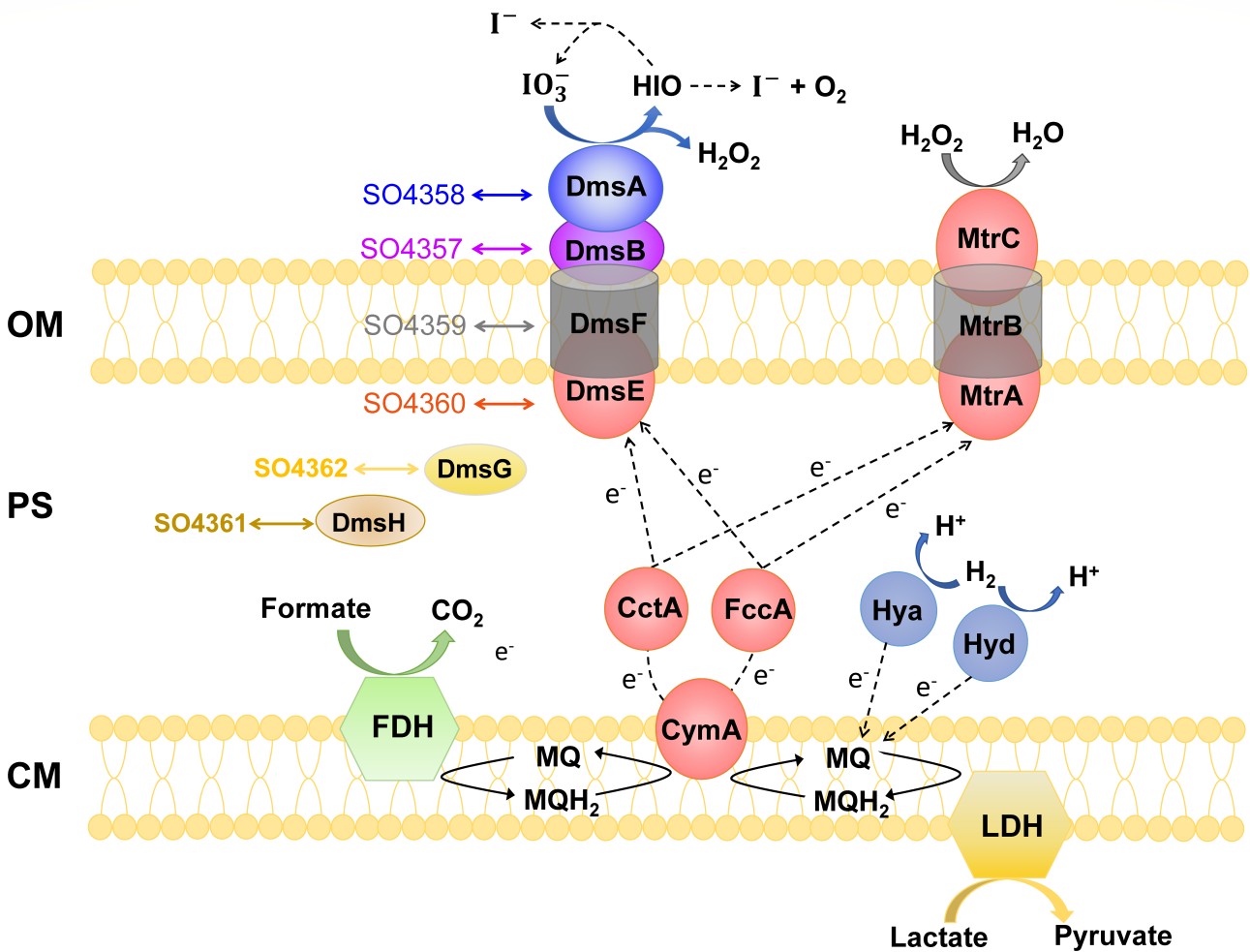

**FIG 5** Schematic representations of extracellular iodate reduction by *S. oneidensis* MR-1 with lactate, formate, and $H_2$ as electron donors and complementation effects between homologous proteins encoded by *so4357–so4362* and *dmsEFABGH* operons. Bidirectional arrows indicate homologous proteins. OM, the outer membrane; PS, the periplasm; CM, the cytoplasmic membrane; MQ, menaquinone; $MQH_2$, menaquinol; FDH, formate dehydrogenase complex (e.g., FdnGHI and FdhXABC) (34); LDH, lactate dehydrogenase (i.e., Dld-II and LldEGF) (59–61); Hyd, hydrogenase complex HydABC (63, 66); Hya, hydrogenase complex HyaABC (64, 65).

This study provides further evidence for the model of $IO_3^-$ reduction by coordination of DmsEFAB and MtrCAB (Fig. 5). Under anaerobic conditions, menaquinone is reduced to menaquinol during oxidation of lactate, formate, and $H_2$. CymA oxidizes menaquinol and transfers electrons to DmsE and MtrA via CctA and FccA. Extracellular $IO_3^-$ is reduced by the DmsEFAB complex with generation of HIO and $H_2O_2$ (26). HIO may be spontaneously disproportionated to $I^-$ and $IO_3^-$ or degraded into $I^-$ and $O_2$ by unknown dismutase (14, 23, 26). MtrCAB complex is involved in the degradation of extracellular $H_2O_2$. As a molybdenum cofactor insertion chaperone, DmsG probably participates in the maturation of DmsA (27). The function of DmsH is still unknown, but it was verified to be involved in extracellular reduction of $IO_3^-$ and DMSO. SO4357–SO4362 cannot form independent extracellular $IO_3^-$ reduction pathway but have complementation effects on their respective homologs among DmsEFABGH.

## Conclusions

This study demonstrated that the *dmsEFABGH* operon encodes an essential electron transfer pathway for extracellular $IO_3^-$ reduction by *S. oneidensis* MR-1 with lactate, formate, or $H_2$ as the sole electron donors. The *dmsG* and *dmsH* genes were verified to be involved in the reduction of both $IO_3^-$ and DMSO. The *so4357–so4362* operon was also involved in $IO_3^-$ reduction through complementing the function of *dmsEFABGH* operon at single-gene level, which indicates the modularity of the Dms pathway. The *mtrCAB* operon was necessary for maximum reduction of $IO_3^-$ with all electron donors due to its involvement in $H_2O_2$ degradation.

## ACKNOWLEDGMENTS

This work was supported by the National Key Research and Development Program of China (No: 2018YFA0901300), National Natural Science Foundation of China (No: 42277065, 42272353), the Natural Science Foundation of Hubei Province (2021CFB214), and the Fundamental Research Funds for the Central Universities, China University of Geosciences (Wuhan) (No: 122-G1323533057).

## AUTHOR AFFILIATIONS

[1]Department of Biological Sciences and Technology, School of Environmental Studies, China University of Geosciences, Wuhan, Hubei, China
[2]State Key Laboratory of Biogeology and Environmental Geology, China University of Geosciences, Wuhan, Hubei, China
[3]Hubei Key Laboratory of Wetland Evolution & Eco-Restoration, Wuhan, Hubei, China

## AUTHOR ORCIDs

Liang Shi http://orcid.org/0000-0003-4546-4741
Yongguang Jiang http://orcid.org/0000-0002-3829-7423

## FUNDING

| Funder | Grant(s) | Author(s) |
| --- | --- | --- |
| MOST | National Key Research and Development Program of China (NKPs) | 2018YFA0901300 | Yongguang Jiang |
| MOST | National Natural Science Foundation of China (NSFC) | 42277065 | Yongguang Jiang |
| MOST | National Natural Science Foundation of China (NSFC) | 42272353 | Liang Shi |

| Funder | Grant(s) | Author(s) |
|---|---|---|
| 湖北省科技厅 \| Natural Science Foundation of Hubei Province (湖北省自然科学基金) | 2021CFB214 | Yidan Hu |
| Fundamental Research Funds for the Central Universities, China University of Geosciences (Wuhan) | 122-G1323533057 | Liang Shi |

## AUTHOR CONTRIBUTIONS

Lingyu Hou, Data curation, Investigation, Methodology, Validation | Beiling Zheng, Investigation, Methodology | Zhou Jiang, Writing – review and editing | Yidan Hu, Writing – review and editing | Liang Shi, Supervision, Writing – review and editing | Yiran Dong, Writing – review and editing | Yongguang Jiang, Conceptualization, Funding acquisition, Project administration, Supervision, Writing – original draft

## ADDITIONAL FILES

The following material is available online.

### Supplemental Material

**Supplemental table and figures (Spectrum00512-24-s0001.docx).** Tables S1; Fig. S1-S7.

### Open Peer Review

**PEER REVIEW HISTORY (review-history.pdf).** An accounting of the reviewer comments and feedback.

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
