## [Reviewer comments · Microbiology Spectrum]

Microbiology Spectrum

The *dmsEFABGH* operon encodes an essential and modular electron-transfer pathway for extracellular iodate reduction by *Shewanella oneidensis* MR-1

Lingyu Hou, Beiling Zheng, Zhou Jiang, Yidan Hu, Liang Shi, Yiran Dong, and Yongguang Jiang

Corresponding Author(s): Yongguang Jiang, China University of Geosciences

Review Timeline:

Submission Date:	February 27, 2024
Editorial Decision:	May 5, 2024
Revision Received:	May 29, 2024
Accepted:	June 3, 2024

Editor: Jing Han

Reviewer(s): The reviewers have opted to remain anonymous.

Transaction Report:

DOI: <https://doi.org/10.1128/spectrum.00512-24>

Re: Spectrum00512-24 (The *dmsEFABGH* operon encodes an essential and modular electron-transfer pathway for extracellular iodate reduction by *Shewanella oneidensis* MR-1)

Dear Prof. Yongguang Jiang:

Thank you for the privilege of reviewing your work. Below you will find my comments, instructions from the Spectrum editorial office, and the reviewer comments.

Revision Guidelines

Sincerely,
Jing Han
Editor
Microbiology Spectrum

Reviewer #1 (Comments for the Author):

The work here focuses on better illuminating the role of the *dms* gene cluster in iodate reduction by *Shewanella oneidensis*. There have been a few papers published recently on this topic, with some discrepancies. The authors generate a series of single

gene deletions and test them with three different electron donors to better understand their role in iodate reduction. Overall the results are clear, and while mostly confirmatory in nature, do help us better understand iodate reduction in *Shewanella*. Two major comments are below, along with some minor comments for the author's consideration.

The experiments seem to be single biological replicates with three measurements. From the figure legends: "The values reported are the means and standard deviations of triplicate measurements." These should be biological replicates to specifically address biological variation rather than technical variation in measurement of iodate. Line 250 reports final growth yields also seem to indicate $n=1$, as no standard deviation for these values are presented. Biological replicates would also help validate the results of the double mutant work shown in Figure 3.

Reviewer 2 raised a question (point 8) regarding negative controls for iodate reduction, specifically looking for a control where electron donor was omitted. This request is especially important considering that the anaerobic work was done primarily in a Coy anaerobic chamber. While the authors claim this is filled with 'pure nitrogen gas' this cannot be the case. To scrub oxygen from the system, Coy chambers require hydrogen gas. *S. oneidensis* will robustly use hydrogen as an electron donor, which may confuse the results here. Considering that the authors state that the electron donor dependence of iodate reduction was one of the major aims of this work, it is strange that controls lacking electron donor are missing. The authors need to do a better job describing their experimental setup such that someone who is interested in the results could repeat them. I don't quite understand how anaerobic tubes were flushed with nitrogen gas while also inside a Coy chamber.

Line 138 - why is this 'novel'? Isn't it simply modified? Please describe why the modification was made here.

Line 146 - what is the length of sequence amplified?

Line 154 - optimized how?

Line 169 - why are you growing anaerobic cultures with excess electron donor? I don't think you need to re-do these experiments, but was there a rationale for only providing half of the required fumarate?

Line 189 - what was the protocol for heat-killing?

Line 206 - growth with lactate and DMSO is also unbalanced (as mentioned above for fumarate).

Line 264 - please be specific - what 'intracellular electron acceptors'?

Line 297 - I don't think the authors can make this statement from the evidence presented here.

Line 298 - what does 'obvious modularity' mean?

Line 315 - Unclear what you are referring to with 'high efficiency'?

Missing references in Table 1

Growth curves in supplemental should be plotted on semi-log scale to accurately depict growth features. See <https://schaechter.asmblog.org/schaechter/2018/07/why-you-must-plot-your-growth-data-on-semi-log-graph-paper.html>

Supplemental figures should be on individual pages with their accompanying figure legends.

Reviewer #3 (Comments for the Author):

This study focuses on the microbially-mediated reduction mechanism of iodate. The authors demonstrated that only *dmsEFABGH* was required for direct reduction of iodate without any dependence on electron donors. Also, the *so4357-so4362* gene cluster of *Shewanella oneidensis* MR-1 was first verified to be active in iodate reduction at single-gene level. In general, this study is straightforward and the methods used are robust. The results of this study are reliable and show new findings in the field.

Detailed comments:

Line 48, microorganism-strain

Line 51, The sentence "These observations --- natural environments" is not supported and should be deleted. The importance of *so4357-4362* function should be added.

Line 69, non-biotic-abiotic

Line 235, the vector itself---the empty vector

Line 246, growth---incubation

Line 295, the word "obvious" is conversational

Line 332, the sentence "Therefore, in addition to DmsE, some unknown periplasmic electron carriers were involved in electron transfer to DmsB" should be placed in the former paragraph (line308-318).

Line 341, To be consistent with gene sequences in the cluster, mtrABC should be mtrCAB.

Line 367, lag---absence

Line 369-392, Figure 5 is a concept graphic and should be described independently. The function of DmsGH should be indicated or described in text.

Response to Reviewers Comments

Reviewer #1:

The work here focuses on better illuminating the role of the *dms* gene cluster in iodate reduction by *Shewanella oneidensis*. There have been a few papers published recently on this topic, with some discrepancies. The authors generate a series of single gene deletions and test them with three different electron donors to better understand their role in iodate reduction. Overall, the results are clear, and while mostly confirmatory in nature, do help us better understand iodate reduction in *Shewanella*. Two major comments are below, along with some minor comments for the author's consideration.

Q1: The experiments seem to be single biological replicates with three measurements. From the figure legends: "The values reported are the means and standard deviations of triplicate measurements." These should be biological replicates to specifically address biological variation rather than technical variation in measurement of iodate. Line 250 reports final growth yields also seem to indicate $n=1$, as no standard deviation for these values are presented. Biological replicates would also help validate the results of the double mutant work shown in Figure 3.

Response: We apologize for this misinterpretation due to inaccuracy in our choice of words. In this study, all experiments were performed in biological triplicate. The means and standard deviations were calculated from the values of triplicate experiments and were shown in the figures. We have changed the phrase "triplicate measurements" into "triplicate experiments". Standard deviations of growth yields were shown in Supplementary Figure S2B. We have added the values of standard deviations in the text as "After 22 hours of incubation, the optical density of $\Delta dmsA$, $\Delta dmsB$, $\Delta dmsE$, $\Delta dmsF$, $\Delta dmsG$, and $\Delta dmsH$ were only 0.21 ± 0.01 , 0.17 ± 0.01 , 0.51 ± 0.03 , 0.26 ± 0.01 , 0.20 ± 0.01 , and 0.25 ± 0.01 , in comparison to 1.26 ± 0.04 for the

wild type (Supplementary Figure S2B).” (line 234-237)

Figure S2 Growth of the wild type and *dmsEFABGH* mutants with DMSO as the sole terminal electron acceptor. (A) Growth over 26 hours. (B) Optical density at 22-hour. WT, *S. oneidensis* MR-1. (A) Growth over 26 hours. (B) Optical density at 22-hour. WT, *S. oneidensis* MR-1. The values reported are the means and standard deviations of triplicate experiments. For points without error bar, the error was smaller than the symbol. Asterisks, significance levels of difference between the mutants and WT, $0.01 < P < 0.05$ (*), $0.001 < P < 0.01$ (**), $P < 0.001$ (***)

Q2: Reviewer 2 raised a question (point 8) regarding negative controls for iodate reduction, specifically looking for a control where electron donor was omitted. This request is especially important considering that the anaerobic work was done primarily in a Coy anaerobic chamber. While the authors claim this is filled with 'pure nitrogen gas' this cannot be the case. To scrub oxygen from the system, Coy chambers require hydrogen gas. *S. oneidensis* will robustly use hydrogen as an electron donor, which may confuse the results here. Considering that the authors state that the electron donor dependence of iodate reduction was one of the major aims of this work, it is strange that controls lacking electron donor are missing. The authors need to do a better job describing their experimental setup such that someone who is interested in the results could repeat them. I don't quite understand how anaerobic tubes were flushed with nitrogen gas while also inside a Coy chamber.

Response:

As suggested by the reviewer, we supplemented a control experiment where

electron donors were omitted. The procedures of medium preparation and bacterial collection and inoculation were the same as previous experiments. IO_3^- reduction was severely inhibited for both wild-type and mutant strains (Figure S1). Although *Shewanella oneidensis* MR-1 may utilize endogenous electron donors, only 21.4% of added IO_3^- was reduced by the wild-type strain and less IO_3^- was reduced by the mutants. However, 93.8% of added IO_3^- was reduced by the wild type with exogenous lactate as the sole electron donor. Therefore, IO_3^- reduction in previous experiments should be ascribed to the added electron donors but not H_2 contamination. We have added this figure and corresponding descriptions in the methods and results of the text.

Figure S1 IO_3^- reduction by *S. oneidensis* MR-1 strains without electron donors. (A) Percentage of added IO_3^- over 36 hours of reduction and (B) Efficiency of IO_3^- reduction at 36-hour. WT, *S. oneidensis* MR-1. One hundred percent of added IO_3^- was equal to 250 μM . The values reported are the means and standard deviations of triplicate experiments. For points without error bar, the error was smaller than the symbol. Asterisks, significance levels of difference between the mutants and WT, $0.01 < P < 0.05$ (*), $0.001 < P < 0.01$ (**), $P < 0.001$ (***)).

In our study, we replaced the mixed gases ($\text{N}_2 : \text{CO}_2 : \text{H}_2 = 95.5 : 3 : 1.5$) in the chamber with pure nitrogen gas to eliminate the contamination of hydrogen gas used for maintaining anaerobic condition of the Coy chamber. Moreover, bacterial inoculation was performed with disposable syringe and there is no need to open the rubber stopper of anaerobic tubes and bottles. We have added these details in the Materials and methods section of the text, including “Afterward, the cells were

inoculated in anaerobic tubes with disposable syringe” (Line 155) and “During bacterial collection, the mixed gases ($N_2 : CO_2 : H_2 = 95.5 : 3 : 1.5$) in the chamber was replaced by pure nitrogen gas”. (Line 161)

To eliminate dissolved oxygen, prepared medium was boiled and purged for 20 min with pure nitrogen gas. These procedures were performed outside the Coy chamber. In the revised version, we have specified the procedures that need to be implemented inside the Coy chamber as “Bacterial washing and inoculation procedures were performed in an anaerobic chamber (Coy Laboratory Products, USA)”. (Line 159-160) We apologize for any misinterpretation.

Q3. Line 138 - why is this 'novel'? Isn't it simply modified? Please describe why the modification was made here.

Response: We have deleted the word 'novel'. The revised sentence is “A suicide vector pDS3.2 was constructed by inserting a short sequence with additional SbfI and AvrII sites in plasmid pDS3.0 (37)” (Line 119-120). Compared to pDS3.0, the pDS3.2 vector provides more restriction sites for construction of recombinant plasmids.

Q4. Line 146 - what is the length of sequence amplified?

Response: The length of each target sequence was added in Table S1.

Q5. Line 154 - optimized how?

Response: The description “optimized ribosomal binding sites” was used to indicate that these sites have been shown to be effective in *Shewanella oneidensis* MR-1. We have modified the sentence as “To complement the mutants, the coding sequence of each gene was amplified by using primers with ribosomal binding sites that have been shown to be effective in *S. oneidensis* MR-1 (40,41)” (Line 134-136).

Q6. Line 169 - why are you growing anaerobic cultures with excess electron donor? I don't think you need to re-do these experiments, but was there a rationale for only providing half of the required fumarate?

Response: We appreciate this reminder. The ratio between the electron donor lactate and the electron acceptor fumarate was not accurately calculated. Considering that a proportion of the electrons released from lactate oxidation will be consumed in anabolism, we usually add excess lactate in anaerobic cultivation of *Shewanella* spp..

Q7. Line 189 - what was the protocol for heat-killing?

Response: The cells were killed by boiling. We have modified the description as “Negative controls were set by adding the same quantity of wild-type cells killed by boiling in the medium”. (Line 170-171)

Q8. Line 206 - growth with lactate and DMSO is also unbalanced (as mentioned above for fumarate).

Response: We appreciate this reminder. The ratio between lactate and DMSO was not accurately calculated. As aforementioned, we usually add excess lactate in anaerobic cultivation of *Shewanella* spp. to provide electrons for both anabolism and catabolism.

Q9. Line 264 - please be specific - what 'intracellular electron acceptors'?

Response: We have replaced “intracellular electron acceptors” with “O₂ and fumarate”.

Q10. Line 297 - I don't think the authors can make this statement from the evidence presented here.

Response: We appreciate this reminder. In previous study, we found that deletion of *so4357–so4360* has no adverse effect on the reduction of DMSO and IO₃⁻ (Guo et al., 2022). Therefore, the *so4357-so4362* gene cluster could not encode an independent electron-transfer pathway towards DMSO and IO₃⁻. We have added this reference in the text “...although *so4357-so4362* could not encode an active IO₃⁻ reduction pathway by itself (26).”(Line 282-283) We also added a description in the introduction section as “However, the role of *so4357–so4362* in extracellular electron transfer has not been systematically investigated. Deletion of *so4357–so4360* has no adverse effect on the reduction of DMSO and IO₃⁻, indicating that the *so4357-so4362* gene cluster could not encode an independent electron-transfer pathway towards DMSO and IO₃⁻ (26).”(Line 81-85)

Reference: “Guo J, Jiang Y, Hu Y, Jiang Z, Dong Y, Shi L. 2022. The roles of DmsEFAB and MtrCAB in extracellular reduction of iodate by *Shewanella oneidensis*

MR-1 with lactate as the sole electron donor. Environ Microbiol 24:5039-5050.”

Q11. Line 298 - what does 'obvious modularity' mean?

Response: In previous study, Coursolle and Gralnick (2010) reported the modularity of the metal-reducing pathway encoded by the *mtr* gene cluster in *S. oneidensis* MR-1. They used the word “modularity” to show the finding that MtrF and MtrD can partially complement the metal-reducing activity of MtrC and MtrA, respectively, which indicates complementation effects of *mtrDEF* gene cluster on *mtrABC* gene cluster at single-gene level. Similarly, our study revealed the complementation effects of *so4357-so4362* gene cluster on *dmsEFABGH* gene cluster at single-gene level. Therefore, we also used “modularity” to show this new finding. The word “obvious” has been deleted as suggested by reviewer 3#.

Reference: Coursolle D, Gralnick JA. 2010. Modularity of the Mtr respiratory pathway of *Shewanella oneidensis* strain MR-1. Mol Microbiol 77:995-1008.

Q12. Line 315 - Unclear what you are referring to with 'high efficiency'?

Response: In this sentence, “high efficiency” was used to indicate that $\Delta dmsE$ reduced more IO_3^- than other single-gene mutants. We have deleted “high efficiency” and modified the sentence as “This result is consistent with the finding that the IO_3^- -reducing efficiency of $\Delta dmsE$ is higher than those of other single-gene mutants”. (Line 300-301)

Q13. Missing references in Table 1

Response: We have added the source or reference of strains and plasmids in Table 1.

Q14. Growth curves in supplemental should be plotted on semi-log scale to accurately depict growth features. See <https://schaechter.asmblog.org/schaechter/2018/07/why-you-must-plot-your-growth-data-on-semi-log-graph-paper.html>

Response: In this study, we used growth curves to determine whether the viability of mutant strains was obviously impaired in comparison to that of the wild type (Figure S2, S3, S7), and whether the viability of double-gene mutants was obviously impaired in comparison to that of single-gene mutants (Figure S4, S6). Statistical analysis was

also performed to confirm significant difference in the values of maximum optical density. For this study, only the values of optical density were available and the number of living cells was not recorded. Therefore, growth curves were plotted as shown in previous studies (Qi et al., *Microbiol. Spectr.* 2024, 12: e02784-23; Xiong et al., *Appl. Environ. Microbiol.* 2020, 83, e01262-17; Yamazaki et al., *Environ. Microbiol.* 2020, 22, 2196–2212). We agree with the opinion of reviewer 1# and Pro. Elio that plotting the log of the number of cells versus time is useful for directly comparing the exponential growth rate of different cultures. We will follow their recommendations in our future work. Many thanks for the reminder.

Q15. Supplemental figures should be on individual pages with their accompanying figure legends.

Response: We have placed the supplemental figures on individual pages with their accompanying figure legends.

Reviewer #3:

This study focuses on the microbially-mediated reduction mechanism of iodate. The authors demonstrated that only *dmsEFABGH* was required for direct reduction of iodate without any dependence on electron donors. Also, the *so4357-so4362* gene cluster of *Shewanella oneidensis* MR-1 was first verified to be active in iodate reduction at single-gene level. In general, this study is straightforward and the methods used are robust. The results of this study are reliable and show new findings in the field.

Detailed comments:

Q1. Line 48, microorganism-strain

Response: The word “microorganism” has been changed into “strain”.

Q2. Line 51, The sentence "These observations --- natural environments" is not supported and should be deleted. The importance of *so4357-4362* function should be added.

Response: To emphasize the importance of *so4357-4362* function, we added a

description as “The *so4357–so4362* operon was first demonstrated to be capable of complementing the function of *dmsEFABGH* at single-gene level”.(Line 32-33)

Q3. Line 69, non-biotic-abiotic

Response: The word “non-biotic” has been changed into “abiotic”.

Q4. Line 235, the vector itself---the empty vector

Response: The phrase “the vector itself” has been changed into “the empty vector”.

Q5. Line 246, growth---incubation

Response: The word “growth” has been changed into “incubation”.

Q6. Line 295, the word "obvious" is conversational

Response: The word “obvious” has been deleted.

Q7. Line 332, the sentence "Therefore, in addition to DmsE, some unknown periplasmic electron carriers were involved in electron transfer to DmsB" should be placed in the former paragraph (line308-318).

Response: This sentence was changed into "Therefore, in addition to DmsE and SO4360, some unknown periplasmic electron carriers were involved in electron transfer to DmsB" and was placed at the end of the former paragraph (Line 308-309).

Q8. Line 341, To be consistent with gene sequences in the cluster, *mtrABC* should be *mtrCAB*.

Response: The gene cluster name “*mtrABC*” has been changed into “*mtrCAB*”.

Q9. Line 367, lag---absence

Response: The word “lag” has been changed into “absence”.

Q10. Line 369-392, Figure 5 is a concept graphic and should be described independently. The function of DmsGH should be indicated or described in text.

Response: An independent paragraph was added to describe Figure 5. The function of DmsG and DmsH was described in the text. The function of DmsH is still unknown, but it was verified to be involved in extracellular reduction of iodate and DMSO.

(Line 381-393)

Re: Spectrum00512-24R1 (The *dmsEFABGH* operon encodes an essential and modular electron-transfer pathway for extracellular iodate reduction by *Shewanella oneidensis* MR-1)

Dear Prof. Yongguang Jiang:

Your manuscript has been accepted, and I am forwarding it to the ASM production staff for publication. Your paper will first be checked to make sure all elements meet the technical requirements. ASM staff will contact you if anything needs to be revised before copyediting and production can begin. Otherwise, you will be notified when your proofs are ready to be viewed.

Sincerely,
Jing Han
Editor
Microbiology Spectrum